# Wood Products for Cultural Uses: Sustaining Native Resilience and Vital Lifeways in Southeast Alaska, USA

Adelaide Johnson [1,*], Audrey E. Clavijo [2], Glenn Hamar [3], Deborah-Aanutein Head [4], Andrew Thoms [5], Wayne Price [6], Arianna Lapke [7], Justin Crotteau [8], Lee K. Cerveny [9], Hailey Wilmer [10], Lillian Petershoare [11], Andrea Cook [12] and Sienna Reid [13]

1 USDA Forest Service, Pacific Northwest Research Station, Land and Watershed Management Program, 11175 Auke Lake Way, Juneau, AK 99801, USA
2 A. Clavijo Consulting, Suwanee, GA 30024, USA; audrey.clavijo@gmail.com
3 Haida Carver, Kasaan, AK 99919, USA; debas127@gmail.com
4 Tlingit and Haida Master Weaver, Craig, AK 99925, USA; aanutein@gmail.com
5 Sitka Conservation Society, Sitka, AK 99835, USA; andrew@sitkawild.org
6 Northwest Coast Art, School of Arts and Sciences, University of Alaska Southeast, Juneau, AK 99801, USA; silvercloud@aptalaska.net
7 Hoonah Indian Association, Hoonah, AK 99829, USA; arianna.lapke@hiatribe.org
8 USDA Forest Service, Rocky Mountain Research Station, USDA Forest Service, 800 E. Beckwith Ave., Missoula, MT 59801, USA; justin.crotteau@usda.gov
9 USDA Forest Service, Pacific Northwest Research Station, Goods, Services and Values Program, Seattle, WA 98103, USA; lee.cerveny@usda.gov
10 USDA Forest Service, Pacific Northwest Research Station, Goods, Services and Values Program, 11175 Auke Lake Way, Juneau, AK 99801, USA; hailey.wilmer@usda.gov
11 Retired USDA Forest Service Tribal Relations Program Manager, Juneau, AK 99801, USA; lvpetershoare@gmail.com
12 School of Arts and Sciences, University of Alaska Southeast Juneau, Juneau, AK 99801, USA; andrea0cook@gmail.com
13 Huxley College of the Environment, Western Washington University, Bellingham, WA 98225, USA; sienna@sitkawild.org
* Correspondence: adelaide.johnson@usda.gov

**Abstract:** Ongoing revitalization of the >5000-year-old tradition of using trees for vital culture and heritage activities including carving and weaving affirms Alaska Native resilience. However, support for these sustained cultural practices is complicated by environmental and political factors. Carving projects typically require western redcedar (*Thuja plicata*) or yellow cedar (*Callitropsis nootkatensis*) trees more than 450 years of age—a tree life stage and growth rate inconsistent with current even-aged forest management strategies. Herein, we qualitatively assess the significance of wood products to rural communities and Indigenous cultures with implications for natural heritage sustainability. In partnership with Alaska Native Tribes, we engaged local youth programs to lead community discussions throughout southeast Alaska to provide specificity to the suite of cultural activities linked to regional forest lands. Results from 58 discussions across 11 southeast Alaska communities (primarily Alaska Native participants) highlighted the cultural importance of forest products including totem poles, dugout canoes, longhouses, woven hats, and woven baskets. Findings indicated spiritual well-being, health, education, tourism, and livelihood significance attributed to these products. Participant-suggested management strategies for increasing supply and expanding access to trees on public lands included: engaging local artisans in forest planning, selecting and delivering specific trees to roads as part of ongoing timber sales, allowing bark removal prior to forest-timber sales, simplifying the tree-acquisition permit process, and setting aside cultural forest groves to sustain trees seven generations into the future. By facilitating discussions, this study fostered relevant place-based youth and community engagement, benefiting youth and enhancing community knowledge transfer while simultaneously summarizing the significance of forest products for resilient culture and heritage activities. Forest management plans aiming to support Alaska Native lifeways may benefit from improved understanding of Indigenous perspectives and worldviews; designation of "culture market values" and "culture targets" can help deliver a broad array of ecosystem services.

**Keywords:** cedar wood products; spiritual and health value; forest sustainability; cultural ecosystem services; citizen science; traditional ecological knowledge; Tlingit; Haida; Tsimshian; environmental justice

## 1. Introduction and Study Areas

As our opening statement, we acknowledge the original Indigenous people who continue to live on their homelands. We would like to express our deepest gratitude to all the southeast Alaska community participants including Tlingit, Haida, and Tsimshian Tribal members who offered their time, wisdom, and perspectives. It is our intention to honor all the southeast Alaska Tribes, elders, youth, carvers, weavers, artists, and stakeholders who participated in this and future Wood Products for Culture and Heritage research. If we as authors have misrepresented the speakers or the data, we apologize. To our readers, we urge you in your work to acknowledge the original Indigenous people of your area [1]

—Gunalchéesh, Háw'aa, T'oyaxsn, Thank you.

The Pacific coastal temperate rainforest of southeast Alaska, USA supports subsistence activities and traditional lifeways for Tlingit, Haida, and Tsimshian Alaska Natives [2,3] (Figure 1). Southeast Alaska has a total population of approximately 75,000 people, including 35 communities comprised of 15% to 88% Alaska Natives [4]. Much of southeast Alaska is within the US federally administered Tongass National Forest, named after the most southern group of Tlingit people, the Taant'a Kwáan, meaning Sea Lion Tribe [5]. High-value wood from old-growth redcedar (*Thuja plicata* Donn ex D. Don), or laaý, ts'úu, ggalaaw in the languages of Tlingit, Haida, Tsimshian, respectively [6–8], and yellow cedar (*Callitropsis nootkatensis* (D. Don) Oerst. ex D.P. Little), xáay, sçahláan, and wahl, in the languages of Tlingit, Haida, Tsimshian, respectively [6–8], has been used from southeast Alaska forests for millennia. Wood and bark used for cultural commodities including totem poles, dugout canoes, longhouses, masks, boxes, bowls, tools, baskets, mats, and hats [9–12]. Along with cedar bark, roots from Sitka spruce (*Picea sitchensis* (Bong.) Carrière), shéiyi, k̲íid, and sha'mn in the languages of Tlingit, Haida, Tsimshian, respectively [6–8] are also used for weaving. Use of western redcedar increased by Alaska Natives [13] with cedar forest expansion following Pleistocene glacier retreat, approximately 6000 years before present [14]. To sustain essential cultural lifeways long into the future, we aimed to engage community members in discussions on wood and bark used for culture and heritage purposes, including the specifics of the forest-product use, significance, and access.

Current distribution, use, and access to cedar trees has likely been altered by colonialism, climate change, and forest management activities that threaten the maintenance of a continually active and robust cultural way of life. Colonialism, including cultural repression, loss of language, and land claims with establishment of a EuroAmerican economy, nearly destroyed Native Americans' traditional ways of living [15–18]. For example, in the 1950s, Native American youth were placed in segregated residential schools with mandates for ethnic assimilation [19]. The resurgence of cultural concerns and demands for sovereignty have become prominent among Native Americans in recent decades [19] and includes repatriating stolen artifacts [20] and renaming sites with original place names [21]. In southeast Alaska, community interest in revitalizing old artforms have included: creation of bentwood boxes, canoe paddles, woven baskets, totem poles, dugout canoes, and clan houses [10,11], culturally responsive teaching [22] such as carving and weaving classes (e.g., University of Alaska Southeast); and creative multimedia using Traditional Ecological Knowledge (TEK) in school programs, e.g., [23]. As described by Pierotti and Wildcat [24], there are two basic TEK concepts: (1) all things are connected, conceptually related to Western community ecology, and (2) all things are related, which changes the emphasis from the human to the ecological community. By describing connections, key to TEK,

integrated conceptual models can be developed and have been used to communicate the interplay between human health and place [25,26].

Geographically, the northernmost extent of western redcedar is between the communities of Klawock and Angoon [27] (Figure 1). Among healthy stands of yellow cedar in southeast Alaska, there are also isolated patches of dying yellow cedar associated with climate change [14,28,29]. Cedar bark and cedar wood used for carving and weaving is obtained from slow-growing centuries-old trees. When bark is taken off the tree correctly for weaving activities, tree growth or survival is not harmed [12,30]. Presence of heartwood chemical compounds, providing a unique aroma and decay resistance to cedar, protect the inner tree [12,31–33]. Incorrect or excessive bark harvest, exposing too much cedar sapwood, can cause lasting damage to the tree.

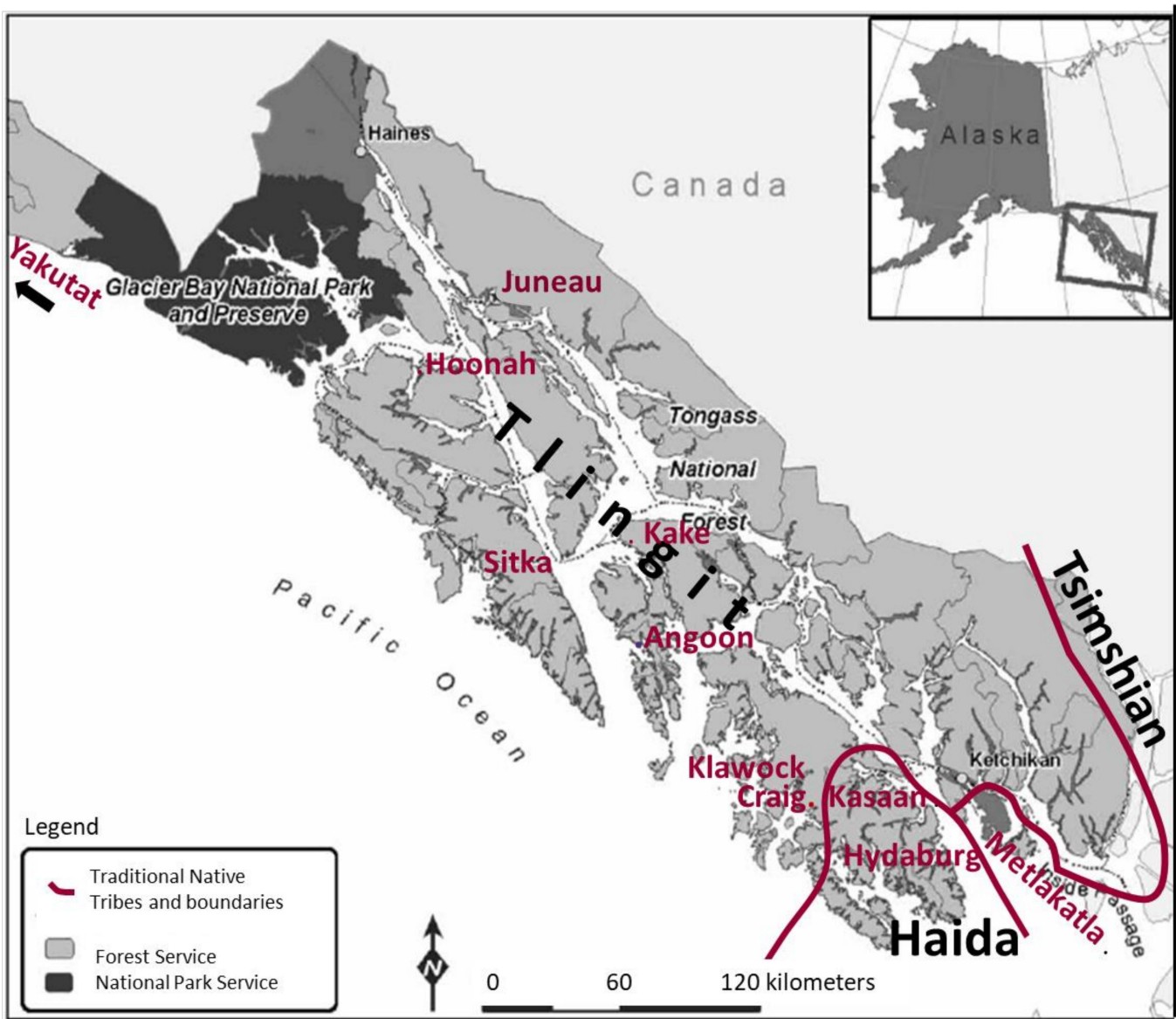

**Figure 1.** Map of southeast Alaska, USA, showing 11 communities with federally recognized Alaska Native Tribes that participated in the study on the Tongass National Forest, administered by the US Department of Agriculture, Forest Service. The Glacier Bay National Park and Preserve is administered by the US Department of Interior, National Park Service. Yakutat, on the coastal northwest, is in the Eyak Alaska Tribal region. Map was adapted from Zagre et al. [34].

Trees used for carving totem poles and dugout canoes are generally over 450 years old, a tree life stage and growth rate not consistent with traditional timber-oriented forest management strategies. Furthermore, tight-grained wood formed by slow-growing trees is best for carving because of consistency and strength. Therefore, old-growth forests (or forests that approximate them) are pivotal for sustaining Native cultural forest use. Young-growth forests are typically managed for fast growth, a trait not conducive to the clear tight growth rings needed to support culture and heritage activities. In addition, even-aged forest management does not harmonize with the Native TEK necessary for sustaining trees "seven generations" into the future [35].

Community cultural significance and threats to access or supply of these culturally important resources warrant systematic assessment in a way that fosters balanced use across all ecosystem products and services. As the dominant land agency in the region, the US Department of Agriculture, Forest Service (USFS) notably and specifically states its "aim to locate and protect areas that contain specific [Tongass National] Forest resources of heritage value used for Native art and craft forms" [36]. As for any shared resource, management for cultural resources includes input and assessment from all ownerships and is improved by collaboration with local community members. For example, in some Australian national parks, cultural heritage has been evaluated in the context of forest planning with a concerted effort for 'joint management', where Indigenous traditional landowners cooperatively manage the park with relevant government agencies [37–39].

To support Indigenous viewpoints and values, Whtye [40] (p. 126) notes: "human and environmental relationships have many possible values including, spirituality, sustainability, sense of place or home, and communion with non-humans." By marking various uses and values of cultural and heritage wood products in this research, we aimed to contribute to a growing body of scholarship advancing the role of forests and parks in sustaining Indigenous access to traditional foods, fiber products [41,42], and ecosystem services related to non-timber forest products [43]. Cultural ecosystem services, including heritage and identity, play a central role in the ecosystem service concepts [44]. Non-timber forest product-based cultural ecosystem services contribute to the social cohesion and expression of Indigenous peoples, linking individuals and communities with a sense of belonging [45]. Systematic assessment of non-timber forest products for culture and heritage can contribute to forest planning and co-management efforts. Novel participatory citizen-science approaches [46,47] can be used to align with data sovereignty considerations [48] while also engaging youth, community members, and managers in meaningful considerations of cultural wood significance in communities. Such youth participation in relevant cultural activities not only enhances Native stewardship values [49], but also is associated with positive health outcomes [50] and advanced education achievements [51,52].

Better knowledge of cultural significance, artist requirements for use of wood and bark, and availability of wood could provide land managers with tools to better assess and plan for Alaska Native needs. While land management agencies are expected to "seek to meet the market demand for Tongass timber" [53,54] among other objectives, the "cultural market demands" also need consideration. Our definition of "cultural market demands" includes recognition of community significance, associated jobs, known income values associated with wood used for cultural artforms in southeast Alaska [55], and community requirements for maintaining robust culture and heritage activities. Incorporation of "cultural market demands" would significantly enhance equity among the categories of public land user groups.

Herein, we report on the first phase of a larger community partnership including Alaska Native community members, US federal agency researchers, youth groups, and non-government organization partners. Our aim is to build capacity, knowledge, and action around wood products for culture and heritage across southeast Alaska by adding specificity to the extent of Alaska Native cultural use, community significance, and cultural resource needs to inform management of cultural ecosystem services, e.g., [29,56–58] to support community resilience [59]. Community resilience is defined here as "the exis-

tence, development, and engagement of community resources by community members to thrive in an environment characterized by change, uncertainty, unpredictability, and surprise" [60].

Specific research objectives were established by our collaborative team as a starting point for future research and programming. These included determination of: (1) specific wood products essential for sustaining cultural lifeways; (2) the array of cultural values associated with wood products including spiritual, health, education, tourism, and income; and (3) specific Alaska Native concerns regarding sustainability and access to forest resources. We sought to address these objectives while simultaneously engaging local youth from Alaska Native communities in research work thereby enhancing cross-cultural communication and introducing a new generation to careers in forest planning, research, and art. Finally, research results are utilized to develop conceptual models of research implications aimed at enhancing stewardship partnerships between Alaska Natives and forest managers. By bringing these wood products and activities to light, this paper brings managers, communities and researchers together in a constructive conversation, creating opportunities to foster partnered stewardship and future collaborations to learn and support community resilience.

## 2. Materials and Methods

Our study started and ended with community engagement (conducted either in-person or remotely) and included multi-partner, multi-tribe interactions, deliberately providing relevant youth engagement activities to facilitate intergenerational cultural knowledge transfer (Figure 2). This methodology was developed through discussions with community members and through discussions with youth leaders working with various youth programs. Our communication plan started with potluck-style meetings with participants contributing various traditional foods with community members in Klawock and Hoonah in 2019. The meetings including carvers, weavers and other interested community members. These meetings were aimed to establish relational trust, listening, and collaboration in a culturally accessible venue. Interests in engaging youth while also enhancing stewardship opportunities for facilitating cultural and heritage community activities were voiced and expanded to other communities. Meetings helped to establish steps to utilize youth programs, engage community members, and gather pertinent information through meetings, data collection and analysis, and sharing again.

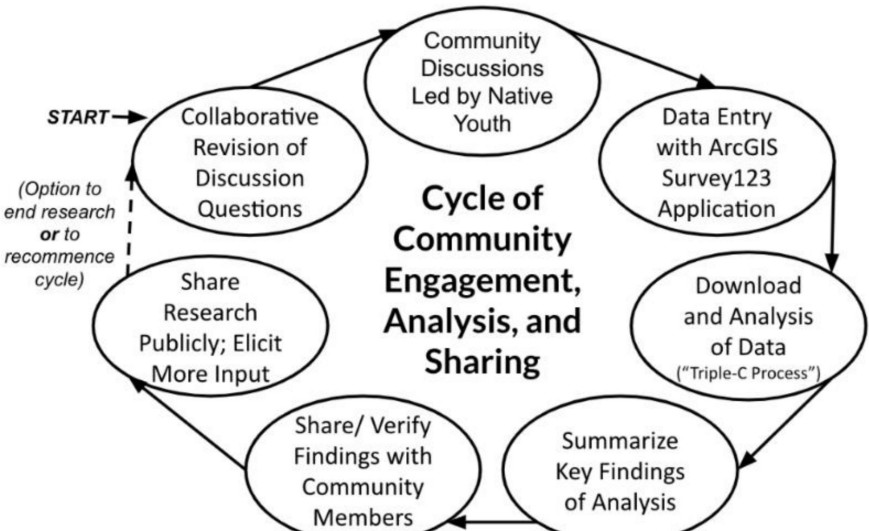

**Figure 2.** Cycle of community engagement, analysis, and sharing. To engage and represent community members, our cycle began and ended with community discussion and data sharing.

## 2.1. Community Discussions and ArcGIS Survey 123 Development

In spring 2019, we held initial community meetings in Klawock and Hoonah, Alaska, to establish goals for the Wood Products for Culture and Heritage project, of which this study is a part of. Project goals included: creating relevant youth engagement opportunities, developing and sharing wood product cultural significance in communities (including spiritual, health, tourism, and income), communicating key concerns about the use of and access to wood products for culture and heritage, and establishing seedling experiments to better understand sites associated with cedar seedling survival. In 2019, eight youth interns in Klawock, Craig, and Hoonah posed an initial set of discussion questions to community members, engaged in field activities with carvers and weavers, and helped to establish a two-year seedling experiment (seedling experiment to be concluded in fall of 2021). In addition, a small mill owner invited weavers to remove bark from yellow cedar trees in a selected harvest site and a tree was selected in the sorting yard for a Hoonah totem pole.

The initial set of community discussion questions deployed in 2019 was further refined in the spring of 2020 with input by carvers, weavers, tribal governments, agencies, and non-government organizations. The Native communities of Kake and Angoon subsequently joined communities of Klawock and Hoonah through development of discussion questions and involvement of youth in the Wood Products for Culture and Heritage project. Community-developed discussion questions included: detailed descriptions of significant wood items in communities; cultural and spiritual well-being, health (particularly with regards to addiction recovery education, tourism, and financial livelihood associated with significant wood items; community member knowledge of the Tongass National Forest's goal to sustain wood for culture and heritage, and questions relating to access, roads, and forest management. Final questions related to messages to and from the community youth (the primary conductors of community discussions). Discussion questions were developed through an iterative process with engagement from multiple Alaska Tribes, youth program leaders, agencies, and non-government organization representation. The Hoonah Indian Association developed an ArcGIS Survey 123 [61] format to lead participants through discussions and record data (Supplementary Materials, Part I).

## 2.2. ArcGIS Survey 123 Deployment

Community discussion questions were posed by 16 youth/young adults from the Alaskan Youth Stewards (AYS) Program and by two university interns, primarily Alaska Native, ranging in age from 14 to 21 years (Figure 3). The AYS Programs included Angoon Youth Conservation Corps (YCC), Hoonah Training Rural Alaskan Youth Leaders and Students (TRAYLS), and Kake TRAYLS/YCC crews. AYS program crews were supervised and mentored by one or more adult leaders in each community (see Supplementary Materials, Part II for further explanation). Discussion leaders followed the Wood Products Project Guide (see Supplementary Materials, Part II), that provided youth with basic interviewing and research procedure training.

All crew members worked with crew leaders or mentors to compile lists of potential interview candidates including weavers, carvers, elders, culture-bearers, USFS personnel, timber mill owners, and tribal representatives within their communities. Community member participants were strategically (not randomly) selected for their expertise or cultural position. Typically, youth contacted and led the discussions with interested community members and recorded responses into the online ArcGIS Survey 123 App. When possible, additional discussions were scheduled with community members as suggested by prior participants.

The two Alaska Native university interns worked primarily with the First Alaskans Institute and Sitka Conservation Society for leadership training to conduct discussions. The interns completed rigorous leadership and cultural sensitivity training prior to conducting community discussions and followed the AYS Program method.

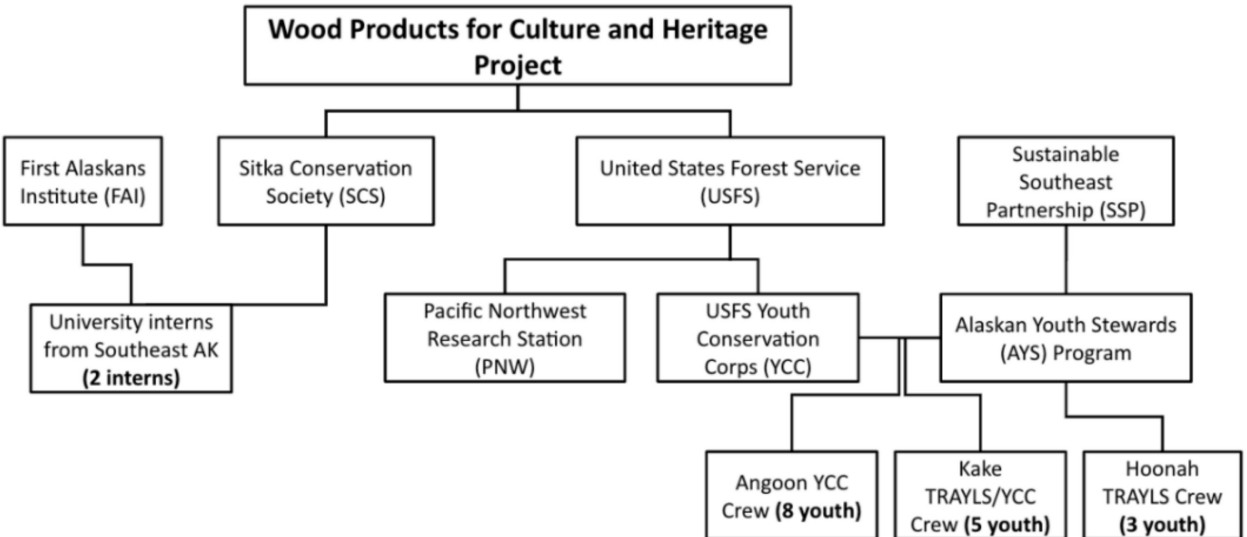

**Figure 3.** Relationship between organizations and youth groups participating in the Wood Products for Culture and Heritage Project. Note: Additional youth program information can be found in Supplementary Materials, Part II.

Discussions were conducted under the "Free and Informed Consent" agreement [62]. Each conversation was recorded (if allowed), transcribed, and paired with a community number (CM#) to maintain participant confidentiality. Respecting Alaska Native data sovereignty, copies of any recorded audio files were delivered to local community tribal archives once the discussions were completed.

Community discussions occurred from July to September 2020 in accordance with COVID-19 public health and safety procedures established by each community.

### 2.3. Data Processing and Analysis

ArcGIS Survey 123 App data were compiled by the Hoonah Indian Association. Survey submissions with corresponding audio files underwent a review and partial transcription and were edited for accuracy and clarity. Quantifiable data, multiple choice, and yes/no questions were sorted and tallied (11 out of 45 questions). Common repeated qualitative concepts were summarized to identify, analyze, organize, describe, and report themes found within the data set [63] in a systematic manner aimed to produce representative results.

From completed community discussions, we developed a "Triple-C process" to code, clump, and categorize dominant community-discussion themes pertinent to study objectives. Two systemized manual coding procedures were used: (1) description-focused coding—describing relevant data and identifying what is portrayed with no judgement or interpretation; and (2) interpretation-focused coding—describing relevant excerpts based on interview questions [64,65]. For example, description-based coding in our data included simple lists of key significant wood products directly reported by community members while interpretation-focused coding included descriptions of management-, health-, or cultural-related benefits associated with carving and weaving. For study objective one, "assess knowledge of specific wood products essential for sustaining cultural lifeways", we used primary codes listing all discussed significant wood products. For objective two, "assess significance of wood products", primary codes included: "cultural/spiritual significance", "health", "education", "tourism", and "income." For objective three, "assess management-related issues", primary codes were "access", "Forest Service", and "Native stewardship." For objective four, "compile youth-related questions", the primary codes included "learn, prepare, engage", "steward of environment", and "connect with and identify with culture". Along with primary codes, there were also secondary and tertiary codes to add specificity and detail to primary codes, making a total of 105 codes (Supplementary Materials, Part III). Multiple codes were used for some data, when appropriate.

A database of all community discussions was reviewed and marked with primary, secondary and tertiary codes. Codes were categorized by sorting parameters pertinent to the four study objectives, then grouped into key themes per study objective. Multiple repeated codes from individual participants were removed to equally represent all participants. For example, if a participant stated four times that greater communication with forest managers was needed, only one coded response was tallied for their four related comments. Categorization was aimed to represent participants as uniformly as possible. Our analysis of key themes was conducted through evaluation of the most frequently recorded code responses per question for all participants. We worked on the premise that repetition was one of the most fundamental ways to identify themes [64,66]. We ranked key themes per study objective by frequency, noting the number of respondents. We acknowledge that these results may not necessarily be in order of community importance, that responses may overlap in scope, that themes may not be all encompassing, and that social dynamics among specific participants in group discussions may have led to elevation of some key themes. Statistical significance of findings was not evaluated considering the non-random participant selection methodology; qualitative assessments of results are provided, enhanced by tallies of common responses per objective and quotations of significant responses. Pertinent quotations, listed by key themes, highlight community perspectives (additional quotes are provided in Supplementary Materials, Part IV).

### 2.4. Data Sharing

As described in our cycle of community engagement (Figure 2), study information will be shared with communities using an online ArcGIS StoryMap [67]. The StoryMap will include photographs, key findings, and videos of interns' relaying their perspectives on community engagement and methodology. It will be presented to federally recognized Alaska Tribes in Hoonah, Angoon, Kake, Sitka, and the Prince of Wales Tribal Stewardship Consortium along with 20 key individuals suggested in multiple community discussions. The StoryMap will be our means of sharing the research study with communities during a peak COVID-19 transmission period with an aim to facilitate future community engagement.

### 3. Results

#### 3.1. Participant Characteristics

A total of 58 youth-led community discussions were conducted in 11 communities (Figure 1). Youth groups in communities of Angoon, Hoonah, and Kake completed a total of 37 discussions while the interns in Sitka and Hydaburg completed a total of 21 discussions (primarily via phone) with community members from an additional eight communities (Metlakatla, Craig, Hydaburg, Juneau, Kasaan, Klawock, Sitka, and Yakutat). Discussions took between one to two hours to complete.

In total, members of Tlingit, Haida, and Tsimshian Tribes comprised 73% of participants and the remaining 27% of participants were non-native (Table 1). Approximately 50% of participants identified gender as female. Age of participants was primarily in the 50–70 year category (45% of participants); all other age categories were <33%. Most participants of the 50–70 age group were affiliated with state or federal agencies and/or tribal associations (81%). For the combined 50–70 and >70 age categories, 62% (34 people) were affiliated with tribal timber corporations and/or mills as compared to 38% in the combined 18–30 and 30–50 age categories (24 people). Out of the collected responses, over 75% of participants had engaged in carving or weaving activities at one point in their lives (78% native, 67% non-native).

**Table 1.** Community discussion participant characteristics. Number of participants listed by tribal identification; gender; age; and any/all past or present affiliations were specified (participants may have multiple or no affiliations). Any carving or weaving experience was indicated. Participants included a subset of available persons, ensuring anonymity.

| Participants | Gender | Age Group | Affiliation | | | Participated in Carving or Weaving? | | |
|---|---|---|---|---|---|---|---|---|
| | | | Tribe | Agency | Mill or Timber Corporation | Yes | No | No Response |
| Native (43) | Women (20) | 18–30 (3) | 1 | 0 | 0 | 1 | 2 | |
| | | 30–50 (6) | 2 | 4 | 0 | 6 | 0 | |
| | | 50–70 (9) | 3 | 1 | 0 | 7 | 1 | 1 |
| | | >70 (2) | 1 | 2 | 0 | 1 | 1 | |
| | Men (23) | 18–30 (2) | 1 | 1 | 2 | 1 | 1 | |
| | | 30–50 (7) | 2 | 3 | 5 | 5 | 1 | 1 |
| | | 50–70 (11) | 5 | 6 | 15 | 9 | 2 | |
| | | >70 (3) | 1 | 1 | 1 | 2 | 1 | |
| Non-native (15) | Women (7) | 18–30 (0) | | | | | | |
| | | 30–50 (3) | 0 | 2 | 0 | 2 | 1 | |
| | | 50–70 (3) | 0 | 2 | 0 | 3 | 0 | |
| | | >70 (1) | 0 | 0 | 0 | 0 | 1 | |
| | Men (8) | 18–30 (0) | | | | | | |
| | | 30–50 (3) | 2 | 1 | 4 | 2 | 1 | |
| | | 50–70 (3) | 1 | 3 | 2 | 2 | 1 | |
| | | >70 (2) | 0 | 0 | 1 | 1 | 1 | |

*3.2. Significant Culture and Heritage Items*

The most frequent (10 of 21) responses to the questions relating to the most meaningful heritage items in communities made of red or yellow cedar were noted (Table 2). Cultural items were described as essential means to share history, ceremony, and traditions. Participants described sacred views of the land; forests not only providing wood and bark for carving and weaving, but also providing a basis for spiritual well-being along with food and other resources needed for survival. Essentially, "the totem pole is like a history book and the screen identifies the family to the clan house, letting everyone know who you are." Traditional uses of some items such as baskets were once considered utilitarian but are now considered either ceremonial or honored household items. Community quotes exemplifying significance of wood products for culture and heritage include:

"Totem poles are incredibly important to us. They are meaningful because it shows that the community's culture is still alive and vibrant."—CM8

"They're meaningful because they keep us in touch with our history and they give our people their identity and purpose for the future. We're identifying ourselves with old things and cedar and we are teaching our young people to carve totem poles, canoes, and the Whale House—not just objects but [teaching them to become] future Haida people. This keeps us alive and gives us a better life too."—CM53

"The most significant ones would be some raven screens and some beaver screens. Át.oow [highly valued clan emblems] carved in the paddles or panels, regalia, Chilkat blankets, which are very valuable and those are made from the bark of yellow cedar. [They are meaningful] because they have to do with our past, our history. A lot of it is, of course, oral history, but it's carried on through the piece. It's important to me because it comes from our culture and the wood comes from our land, our sea."—CM13

**Table 2.** Most often described significant community wood or bark culture and heritage items. Item significance descriptions are taken from participant responses (number in parentheses represents number of participants reporting out of a total of 58 discussions).

| Heritage Item Type | Significance |
|---|---|
| Totem and mortuary poles (26) | Describe 'world views, and history' of a place/person/clan; totems represent Native peoples' 'relationship with nature' and 'protection of sacred lands'. 'Mortuary poles hold the ashes of loved ones' who have passed. |
| Woven cedar bark or wooden hats (17) | 'Current ceremonial uses', 'previously used as a 'utilitarian item' (woven hats) or 'for regalia (wooden hats)' [1] |
| Dugout canoes (11) | Traditional means of travel for harvesting, fishing, and bartering. 'There used to be thousands of dugout canoes down the northwest coast' and some communities have none. Now, they are often carved for 'canoe journeys [which] help promote and perpetuate the cultural connection to the wood and carving as a craft' or 'as part of alcohol and drug recoveryprograms'. |
| Cedar canoe paddles (9) | Typically, one of the first pieces a new carver works on as an 'introduction to carving and basic Formline' for youth at culture camps, schools, and universities. Often display one's 'clan crest'. |
| Woven baskets (7) | Traditionally a 'utilitarian basket' woven 'to pack things like berries and water'; baskets are made from cedar bark and/or spruce roots and passed down through families or given as gifts. |
| Cedar panels/screens (9) | 'Carved panel [of wood] inside or outside of a building' (such as a home/longhouse); typically display clan crests or depict a story/legend. |
| Traditional regalia (5) | Cedar is used for numerous regalia pieces including necklaces, earrings, rattles, rope, leggings, headbands, and sashes; also 'woven into fabric' in Raven's Tail and Chilkat weavings. |
| Cedar masks (5) | 'Traditionally used for armor' (gifted via ceremony for the protection of the wearer); carved masks hold a spiritual and ceremonial value. |
| Chilkat/Ravenstail weavings (5) | Chilkat and Ravenstail weaving uses a loom. The warp is a blend of mountain goat wool and cedar bark. 'Chilkat blankets [were sometimes traded] for [totem] poles.' |
| Tribal/clan longhouse (4) | Longhouses made of cedar (traditionally the home for each clan) with ornate cedar house poles and screens/panels that tell the history of the clan. Many community members would like to see tribal longhouses 'rebuilt or restored'. |

[1] Hats woven from spruce roots are common in Hoonah and Angoon, where cedar trees are less prevalent.

### 3.3. Description of Wood Product Significance

Responses to the question, "How do you think carving, weaving, artistic events/traditional activities (pole raising, canoe journeys, carving shed open houses, classes and apprenticeships, etc.) provide to the community's cultural/spiritual well-being, mental and social health/wellness?" were categorized. Broad themes included cultural and spiritual well-being, health, education, tourism, and livelihood (Table 3). Cultural item value or activity value associated with the item were often considered "priceless". Cultural/spiritual significance is exemplified by the following statement:

> "When I was a kid, I didn't see it as often as I do now. There used to only be certain people who had woven hats and carved masks. Right now, we are in the renaissance of our art. I think it's because our people are tired of conforming to this society and we are proud of our art."—CM33

For health significance of wood products for culture and heritage, most participants described health recovery associated with participation in cultural activities and use of art as an alternative to substance abuse. People in multiple communities described depression associated with historical trauma and the recovery from historical trauma with "the healing canoe" or "healing totems"; their discussions as described by statements including:

> "I think that drug and alcohol abuse here are directly linked to the historical trauma that's occurred through this community being taken over and changed. And not just this community but communities throughout southeast Alaska have been heavily impacted by colonization. All those things just add to the

trauma that's been experienced through colonization and I think that carving and weaving are an important part of healing from that."—CM1

"[They are depressed and cultural activities are a healthy outlet.] When they're trying to clean themselves up, we look for a way to reach them. For our people, the greatest tool is through culture: through the arts, through the language, through the dance. Through the artform, it offers a way for you to come out of it. To start thinking clearly and building a sense of pride and understanding in who you are and where you come from."—CM14

"I think that [carving/weaving] significantly contribute to the cultural, physical, and mental well-being of the community. And it provides an opportunity for uniting across cultures and generations as well. The process of creating is very healing because it connects you with so many lessons that are so much deeper than the actual act of creating something."—CM5

**Table 3.** Community significance attributed to wood products for culture and heritage (number in parentheses represents number of participants reporting out of a total of 58 discussions).

| Significance Attributes | Specific Description of Significance |
|---|---|
| Cultural/spiritual well-being | Learning from elders, passing on knowledge (48) <br> Cultural knowledge/literacy revitalization (40), loss of cultural knowledge (13) <br> Language (7) <br> Native values (32) <br> Traditions, rituals, and ceremonies (28) <br> Identity (28) <br> Connection to: community (23), culture/heritage (21), nature (10), universal connection (9), interwoven (7) <br> Frame of mind: associated with joy, peace, feeling good, feelings associated with a better time (16) <br> Sharing stories, songs (13) <br> Creativity (7) |
| Health (particularly drug and alcohol-related) | Health recovery (50) <br> Regain cultural/spiritual well-being (31) <br> Use art as an alternative to substance abuse (22) <br> Keep clean body and mind (11) <br> Help deal with trauma (10) <br> Help deal with colonialism impacts (8) <br> Health, general well-being (6) |
| Education significance | Importance of culture classes to pass on traditions (41) <br> Recognition of necessary investment, expense (41) <br> Interest in maintaining school (38) <br> Apprentice opportunities (21) <br> Grants/funding opportunities (15) <br> Carving shed for education (13) <br> Culture camp for education (11) |
| Tourism | Increase opportunities (33) <br> Sharing with visitors (25) [1] <br> Commoditization/Commodification (5) <br> Cheaply made, inauthentic replicates (5) |
| Financial livelihood | Income opportunity (22) <br> Not financially important/viable (13) <br> Carving and weaving-related activities as very important for income (31) [2] <br> Barter (7), don't want to sell (5), capitalism (2) |

[1] Tourist activities directly associated with wood or bark including tours of a local carving shed, totem park, or mill, weaving and carving demonstrations, selling artwork to tourists, and/or proprietorship of a store that sells local wood products. [2] Income-related activities.

Educational themes brought up in community discussions included descriptions of school programs, culture camps, and carving sheds as important ways to share practices of carving and weaving. Culture camps are either day or overnight camps where youth are taught traditional skills including preparation and preservation of cultural foods, practicing cultural artforms, participation in traditional singing and dance, and sharing with elders in communities. Carving sheds are buildings or shelters where experienced carvers work on community projects, teach community members, and are often places where carvers engage with travelers and tourists. Carving training typically includes learning Formline with drawing, painting, and carving. Formline is considered, "a natural order created through three universal forms including ovoids, U-shapes, and S-shapes. Formline is a key style of communication uniform amongst Tlingit, Tsimshian, and Haida people." When asked how additional funding could be used, most participants described the need for materials (i.e., cedar wood and bark, which are becoming increasingly expensive) and funding to "compensat[e] artists for their knowledge and time." Many others stated the need for funding to "facilitate more workshops and classes," apprenticeships, and other ongoing art and culture programs in the schools and communities. Community members said:

> "Since a lot of our elders that were raised traditionally have passed on, we're getting a disconnect in our culture. So, the learning about our culture is [now starting to be] taught through schools [instead of being learned] at home. It has created a disconnect between the youth and the older people in our culture. So, I think activities like canoe journeys, raising new poles, all those kinds of things are ways that the younger generation can find connection with our culture."—CM12

> "As far as the carving and Formline design, just to pass on the tradition of what was passed on to me, what was passed on to my father from the generations before, is continuing to show love for the traditions and the ancestors before us... and the love for our traditional arts. It's definitely very important."—CM29

Approximately half of the study participants reported being involved in tourism and about a quarter of these participants (16) indicated being involved in a tourism activity that had a direct relation to wood products. Wood product-related tourism included tours (e.g., of a carving shed or local mill), demonstrations, and the sale of wood products to visitors. Some participants also reported issues with commodification and the sale of false and cheaply made, inauthentic replicates as a barrier to opportunities for local artisans. Inauthentic duplication is a form of cultural appropriation. Community members discussed how tourism increases opportunities to sell products and spread interest as well as the ability to share the culture and art with visitors as exemplified by the statement:

> "Tourism provides an outlet for artists to sell and to celebrate arts. By having outside visitors, it provides that market and the access to the products."—CM5

More than half of participants indicated that carving or weaving (also collection and bartering) was very important to household income. In some cases, bartering of food items with ceremonial objects are described as being 'key to survival':

> "The reason there were so many canoes, totem poles, baskets, jewelry, and clothing is because there was a demand and their living could be supplemented or traded (for goods or US currency). In my current situation, my wallet gets thinner and thinner. People struggle to scratch out a living. The most sustainable is to improve the community (in order to create sustainable tourism). Delivering resources (money) with tourism. I have grown children and grandchildren and within my own life I have appreciated being able to have my lifestyle—being part of this type of culture. I want, for all the people that come after me, I want them to have the option [to have this lifestyle/be a carver] as well."—CM53

> "I have carved a lot . . . a lot of cedar! Hats, masks, totem poles, miniature canoes, paddles, boxes, panels, and screens. I've made my living off of carving for the last 20 years. It affords me to be my own boss. I was able to raise my first son doing it... It's just been a part of my life for a very long time."—CM33

### 3.4. Forest Management-Related Concerns

The efforts of local management (especially per USFS Tongass Land and Resource Management Plan to sustain wood for culture and heritage) was known to 50% of the 58 participants, and awareness was generally well-distributed among age, affiliation with agencies or Alaska Tribes, and carving or weaving experience. Of the five participants approving of local forest management for culture and heritage activities, three were associated with forest harvest activities or lumber mills. Themes associated with management include access, communication, and respect for culture and heritage use of trees (Table 4).

**Table 4.** Management pertaining to agency and Alaska Native practices of sustaining cultural trees (number in parentheses represents number of participants reporting out of a total of 58 discussions).

| Maintaining Trees for Culture and Heritage | Description |
| --- | --- |
| Agency management-related themes | Access opportunities/decrease access barriers (53) <br> Communication and collaboration (51) <br> Roads on public lands (45) <br> Honor wood products for culture and heritage (43) <br> Challenges navigating agency permitting process (39) <br> Forest harvest (20) <br> More engagement of agency with cultural activities (16) <br> Place more value on trees, currently undervalued (10) <br> Increase access opportunities/decrease access barriers (53) <br> Communication and collaboration (51) <br> Road management (45) <br> Honor wood products for culture and heritage (43) <br> Permit process (39) <br> Forest harvest (20) <br> Engage in culture (16) <br> Place more value on trees, currently undervalued (10) |
| Native practices for sustaining trees themes | Stewardship activities (36) <br> Practices including respect, reciprocity, balance, gratitude (30) <br> Selective tree harvest (24) <br> Learning from elders, passing on knowledge (15) <br> Connection to trees (11) <br> Traditions (8) |

#### 3.4.1. Access, Roads, Timber Harvest, and Sustaining Trees for Culture and Heritage

Community members described roads as an important means of accessing cultural resources (i.e., wood and bark) although access/roads were also described as "a double-edged sword", "a mix of benefits and downfalls", and "[a] danger to our people." "It has to be a balance because roads can really be destructive. But roads can also provide access." Additionally, participants described that new tree harvests typically increase distance to potential culture trees for harvest as exemplified by the following statement:

> "All the logging has created all the roads to go get it, which is nice. But it also diminished the quantity of cedar so it's a double-edged sword. They gave us the roads to get to it but, they also cut and sold most of it."—CM29

Participants shared concerns that trees were not being sustained for culture and heritage and identified three main barriers to access wood products for artists and community members: timber harvest, bureaucratic processes, and road access.

1.  Timber harvest: Timber harvest (logging) was described as the main impediment for access to trees for culture/heritage. Participants expressed that "logging had greatly decreased the quantity of tree resources." One community member stated, "The use of wood has become more challenging for many people due to extensive logging. The wood is just more scarce." Other community members mentioned the

effects of logging on the ecosystem and culture lifestyle, stating, "When you destroy a forest, all the animals and streams will be affected by it and it breaks down our way of life." Clear-cut plots and young-growth stands were described as difficult to traverse, making access to the remaining trees beyond the clear-cuts extremely arduous, inefficient, and expensive. Some participants expressed the desire to "cease all harvesting by logging companies". Others suggested calling for a "change in local land management policy, to increase the conservation and sustainability of resources in order to decrease rates of resource extraction."

> "If you change the road system, it's usually to access more timber for clear-cutting, which would lead to less trees for cultural use. The roads aren't built for that (cultural) purpose. They're built for accessing large stands of timber for commercial extractions."—CM21

2.  Bureaucratic processes: Participants shared frustrations with the current administrative permit process (to secure trees for culture use on USFS lands) policies. These include: (a) the "complexity of the permit process" and "lack of available [land management] assistance", (b) the inability of individuals to apply for the permit (an organization such as a Tribe must apply for a permit), and (c) the sentiment that Alaska Natives should not be required to apply for a permit to harvest cultural-use resources, and (d) a lack of direct communication with land managers. The local permitting process and/or policy complications lead many community members to rely on alternative sources (e.g., sawmills) to access wood/bark for culture and heritage. Five out of 58 participants said that the permit process was not a problem.

> "I didn't know that there were rules about getting a lot of these things... I'm not even sure if I know them right now (rules regarding gathering roots). It's taken me, off and on, 10 years to get a tree for the totem pole by my house for my clan. I had to get corporations involved to help me. The USFS policy was impossible because they said I would have to go [from a northern community] to Ketchikan and find my own tree. Also, I had to be denied by many of our corporations to get the 'okay' from USFS. It was defeating."—CM12

3.  Road access: Community members included the closure of USFS logging roads as an obstacle. Many participants expressed that although logging roads may increase timber extraction by logging companies, they are also "important for access [of forest resources]" for local people, "especially in rural communities" and the closure of existing roads "would be very restrictive." Community members voiced the concern that road closures would cause "areas that would've been accessible to no longer [be available]" and that more expensive means of tree removal (i.e., use of helicopter or boat) would be required. It was suggested that land managers could increase community resilience by designating some road networks for cultural and recreational purposes, exclusively.

> "Less roads, closed roads, means less access to roadside wood and necessitates helicopter use. More roads could offer more access, but I don't think anyone has the money to make a road for cultural wood. While managing roads, we do have to create access. It has to be on them [the Forest Service] to mandate however many logs per year and let them do the most efficient way."—CM57

### 3.4.2. Native Approaches

To improve forest health and resource access, community members suggested that forest managers adopt Alaska Native approaches for sustaining trees by increasing the use of stewardship practices. Participants, when describing further detail of what stewardship meant to them, included the terms: respect, reciprocity, balance, and gratitude (Table 4). The stewardship practices most often described were the practice of using limited selective

tree harvest ("never taking more than you need") and the practice of thanking or honoring the tree before or during harvest. These values and traditions are passed down through community elders.

> "Historically, the rate of tree harvest was low enough to pull trees out sustainably. In modern times, the rate of use is great with demands for multiple products, shipped to faraway places. Old growth is a much higher quality and therefore important for culture and heritage projects. We've identified patches of cedar (and other woods), and we've refrained from clear-cutting, and we've selectively taken for local use. With the clear-cuts that have already occurred, I would shy away from the thinning programs as they are currently because it allows for too much light and the trees grow with limbs and knots and wide growth rings (weak with low resilience to rot). This type of timber is not good for cultural use. Let clear-cuts grow up without manipulation. Also, some clear-cuts, may be places where cedar grew well before—areas that are not overly exposed to wind or snow. Snow weighs down the top, eventually breaking it off and then facilitates decay. The straightest trees grow on the north side of a hill in flat, low spots (in a ravine)."—CM53

> "We have provided for a number of totem poles over the years (many yellow cedar, a few redcedar). Of course, that means for us (the mill) to harvest them [for the community]. Finding the right tree, asking the community members' opinions on the selection, handling it gently, getting it to the mill, etc. A lot of work goes into it."—CM3

### 3.4.3. Opportunities to Increase Trust and Celebrate Wood Products for Culture and Heritage

Participants had six main suggestions for increasing trust and celebrating wood products for culture and heritage. These were: (1) increased communication with land managers, greater awareness of the agency's policies, regulations, permit process, and harvesting opportunities; (2) more engagement of land managers in cultural activities and artforms in order to gain hands-on experience, appreciation, and a deeper understanding of these cultural uses of wood/bark; (3) greater participation of land managers in workshops to increase communication and trust between agency officials and community members/artisans; (4) more interactions with artists to share forest knowledge with land managers to create a better understanding of what artists are looking for in trees to be used in culture/heritage projects; (5) commission of local artists to make pieces to be either displayed in the community and district office(s) and/or sold; and (6) ideas of ways that access to wood and bark could be improved.

The most frequently recorded suggestion for increasing access was to have agency officials identify trees that could be used for culture and heritage activities and to communicate the location of these resources and opportunities for harvest to community members. One participant suggested that land managers and mill operators could communicate to community members about the timing of timber sales so that local weavers could collect the bark before the trees are harvested. That said, it is important to note that bark collection is seasonal and typically occurs in the spring or early summer when the sap in trees begins to flow. Bark, not often used by local lumber mills, is discarded. Several participants recommended that the Forest Service and/or private mills maintain a stockpile or reserve of cultural-use trees would greatly increase access to wood. For example, land managers could donate a predetermined number or percentage of trees each year per timber sale to be cut (harvested) for community artisans who have taken a role in selecting specific trees. Other suggestions for improving resource access included the provision of tools, transport, and funding to harvest resources, increased collaboration, increased tribal management opportunities, and the planting of trees. Specific quotes about road access, cooperation with local mills, forest sustainability, and opportunities for national forest management to support culture and heritage activities include:

"[Land managers can enhance opportunities for obtaining wood products by] incorporating indigenous perspectives and worldviews into shaping policy so that access is a reality and so that there is a continuation of these traditions and artforms and the positive social effects that come from practicing the artforms. Having cultural perspectives considered when shaping policy is really essential to ensuring that you're not prohibiting and being a barrier. It's taking into consideration viewpoints that you maybe didn't think of before."—CM5

"For weavers, if they are going to commercially harvest cedar trees, will they let us first strip the bark off the trees? And then I can take 100% of the bark because they are going to cut it down anyway. For the trees [for carvers], do we have identified trees that we can take at a later date? We need to save them and take them as we have a need for them." (The trees will rot if you cut them down too long before you can use them).—CM2

"I think working directly with communities and those who have an interest in maintaining those items (cedar bark is huge for many projects) and dialogue back and forth to educate from both perspectives what kind of management has to take place to make sure that it's not devastated and gone forever. So, how much can you take and still be sustainable? And how much needs to stay so that future generations will be able to even know that it existed? That takes science. It takes studying how things have been in the past and how they are now. The kinds of things that not only cutting down trees but the kinds of environmental things that change the ability for the forest to thrive. [Land managers have] a big responsibility and a big opportunity to utilize those things in a positive way. And serving the forest in a productive and healthy and respectful way serves everybody."—CM46

### 3.5. Messages to and from Youth and Interns

For the question, "What message would you like to convey to youth?", there were three common themes relating to: (1) learning, preparing, and engaging in the artforms and skills; (2) being a steward of the environment; and (3) connecting and identifying with the Native culture (Table 5). Statements indicating community support for engaging youth are exemplified by the following statement:

"More young people are finding opportunities to become artists and seeing they can have a culturally meaningful life for them and their families. It's becoming increasingly important as a refuge and alternative to drugs."—CM21

The community discussions provided space for youth researchers to add any comments and/or reflections at the completion of each community member discussion. Common messages from youth and student interns about their discussion experiences included gaining new knowledge or perspectives, feeling a desire to learn, prepare, or engage more deeply in their culture and/or heritage, and general reactions regarding their role in conducting the survey. Additionally, community conversations were personally meaningful or significant as exemplified by the statements:

"Educating and supporting [the youth] is the best way to make sure these traditions get passed on. Every community needs a space for Indigenous culture sharing. In Sitka, that place seemed to be the culture center which no longer exists. Then again, it's really the people in those spaces, which make them so valuable. We need to support [youth] in the community so that they one day feel they can take on these leadership, mentorship roles."—Youth Researcher

"Our ancient artforms require straight grain, old growth red and yellow cedar. Meaning our cultural expression is at stake because we use our art to represent our families, stories, and histories. That said, the land is occupied by non-natives, too. I think that everyone who breathes the crisp air of the Tongass or eats fresh

salmon from these rivers should consider doing their part in voicing that the land should remain protected."—Youth Researcher

"Through the interviews I learned that a lot of people suggested, and now I think [also], to get forest officials involved in these cultural practices that are so, so important to Native cultures in our region. Some people suggested going out and gathering cedar bark with weavers, looking for the specific conditions you need for a totem tree or a canoe tree. If we know what we are looking for, we can better protect them and save them for Native access. And for carving, getting people out into a carving class with a carver [firsthand], I think that would make a huge difference in forest officials' understanding in the importance... It's just hugely important to be involved, and be aware and informed, and to care about it."—Youth Researcher

**Table 5.** Community members' messages to the youth and youth messages/reflections post discussions (number in parentheses represents number of participants reporting out of a total of 58 discussions).

| Messages/Reflections | Description |
|---|---|
| To Youth | Learn, prepare, and engage with these traditional art and harvesting skills from elders or others; seek out someone to teach you (33)<br>Be a steward of the land/forest; protect and sustain our natural resources (18)<br>Connect with and identify with culture (11)<br>Be an advocate for what you want through management and decision-making processes (7)<br>Create opportunities for youth, pass on traditions, serve community, and learn about history and identity (4) |
| From Youth (post-discussion) | New knowledge, perspectives about wood products, tree access, cultural traditions (23)<br>Learn, engage, prepare with community cultural traditions (10)<br>General statements about the interview experience such as, "I liked the stories", "fast pace", but "did not like redundant questions" (10)<br>Community member and/or responses were meaningful, inspiring (6) |

## 4. Discussion

Our summary of southeast Alaska wood-product community significance along with specific community suggestions for fostering greater access for culture and heritage wood and bark, provides a first step towards: (1) recognizing cultural ecosystem services associated with wood products as promoted through relevant youth-led community engagements, (2) conceptualizing integrated socio-environmental factors supporting cultural/community resilience through identification of "culture targets", and (3) developing management approaches aiming to increase communication, access, and sustainability of wood products for culture and heritage. Our research also sets the groundwork for pertinent future research directions.

### 4.1. Community Engagement Led by Youth to Understand Significance of Wood Products for Culture and Heritage

By conducting discussions, youth socially connected with elders, artists, and others, including agency employees, mill owners, and non-government workers. Through the research process, youth became more aware of the interdependent interactions between culture, forest stewardship, and community health. Like other citizen science youth engagement studies, e.g., [68], this study inspired youth to be greater integrated stewards of the land and their culture. The engagement of youth in relevant community projects both resonated with community members and appeared to provide deeper and more culturally relevant conversations than possible with researchers visiting from outside communities not directly affiliated directly with Alaska Tribes. Continued support for such programs may help ensure that the activities and art forms continue promoting community health and community resilience [69]. In particular, student activities provide a powerful communication link between land-management agencies and local communities. Students

leading discussions gained and shared knowledge of understanding of ecosystem services provided by forest non-timber products [43–45].

Our study provided two key opportunities for Alaska Native youth, the cultural and environmental stewards of the future: (1) relevant experience with forest management work needed to sustain cultural practices into the future and (2) healthy work habits leading to furthering educational pursuits. As found in this study, Alaska Native youth are especially important in maintaining and improving cultural and individual resilience in Native communities by engaging with elders and sharing community traditions. Like other research studies, our youth-led community discussions highlighted connections between cultural identity and traditions that reduce likelihood or persistence of substance abuse in Native American youth, e.g., [50]. Development of opportunities for Alaska Native students to gain skills, knowledge, and experience in cultural and scientific fields has been shown to help Alaska Native students graduate from high school and continue to college [51,52]. Incorporating ' culturally responsive schooling' may greatly "[improve] the education and [increase] academic achievement of American Indian and Alaska Native students" [49,51,52]. Further support of Alaska Native students in pursuit of higher education may effectively lead to culturally diverse management agencies (e.g., USFS) and aid in stewardship of Native values and culture [70–72]. Participants were able to perpetuate Alaska Native knowledge, local culture and values by sharing their perspectives, passions, and stories with local youth, and in some cases, furthered passion of youth and interns in vocations including art and ecological management.

Future conservation initiatives with youth including seedling experiments, silviculture workshops, and timber plot surveys can be used to assess site conditions best promoting survival and growth of red and yellow cedar. Such hands-on field experience may inspire Alaska Native youth to pursue careers within the land management agencies, thereby improving the agencies' diversity and capacity to represent Native values and perspectives. Further research is needed to uncover the best means (e.g., youth development programming, language and traditional skills trainings, etc.) by which cultural revitalization can be achieved and how these programs may influence multiple socioeconomic trends.

*4.2. Conceptualizing What Is Needed to Foster Community Resilience and Identify Culture Targets*

This study's community discussions, illuminating key themes associated with wood products for culture and heritage, directly link forest use to human well-being and community resilience. Repeated themes included "connecting youth with elders", "healthy alternatives for youth", "spiritual/cultural wellness" including "balance", "healthy lifestyles", "relationships with elders, nature, and community", "tradition", and "forest stewardship"; connections key to traditional ecological knowledge (TEK) [24]. These connections, along with access to resources helped foster community resilience [59,60]. Youth and community participation in cultural activities supports healthy lifestyles and community resilience. Like other Indigenous Alkan studies using integrative health models [25], we adapted the One Health model [26] to show physical, biological, and human connections to conceptualize community environments having full access to tree resources providing essential culture and heritage activities (Figure 4a). Participation in community cultural activities required availability and access to cedar trees used for cultural and heritage items. Factors including high cost of trees, distance to pristine forests, and permit processes were described by community members as limitations to tree access. Where resources supporting cultural activities are lacking due to low accessibility of high value trees, the reduced biological (forest) environment limits wood products necessary for culture and heritage activities (Figure 4b).

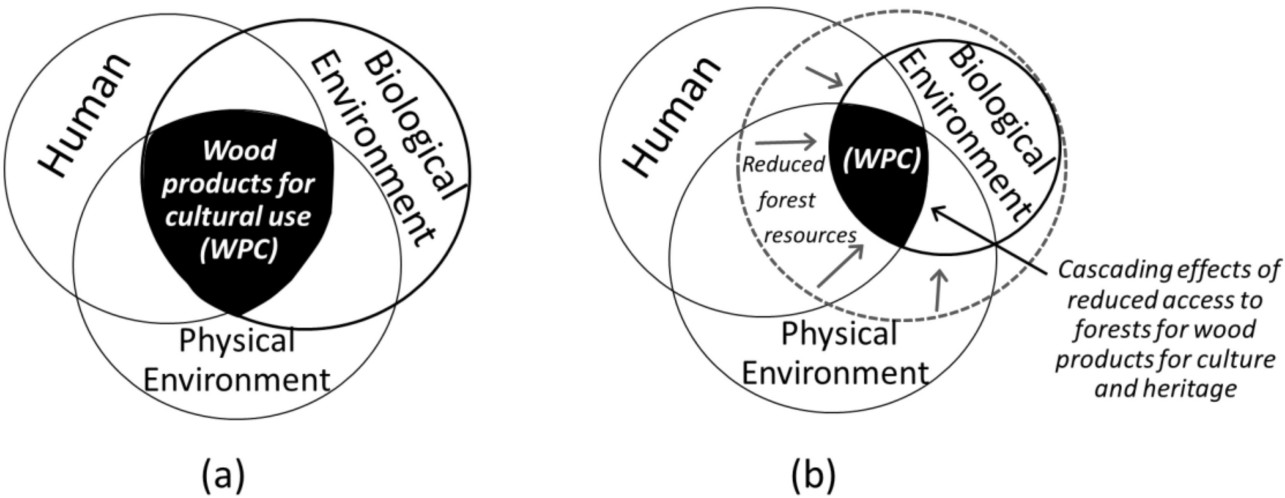

**Figure 4.** Model for a community suggested "culture target" that integrates the physical environment (geology, landforms, streams, shorelines), biological environment (forests and all wild species), and humans for systems: (**a**) at full capacity for providing all vital culture and heritage activities and (**b**) at reduced capacity.

To facilitate greater access to trees supporting cultural use of wood products, community members described a need for sharing a specific community's requirement, or "culture target". A culture target would include economic demand calculations ("culture market values") for cultural and Pacific Northwest carving arts along with the wood product demand needed to sustain this sector [55]. We define a "culture target" as both the short-term and long-term requirements for sustaining forests for wood products for culture and heritage activities seven generations into the future [35]. It was not in the scope of this project to describe specific culture targets. Five quantifiable components suggested in discussions to increase community members' access to cultural resources included: (1) provide trees to carvers and bark to weavers, as part of the ongoing timber sale process, (2) identify and map locations of culture and heritage reserves of cedar trees that would be held for sustainable cultural use, (3) withhold silvicultural actions in some young-growth stands to promote slow growth of cedar, (4) plant cedar, and (5) provide opportunities for youth to engage in stewardship activities. Future work is needed to address community culture targets and culture market values fostering community resilience, cultural/spiritual well-being, health, education, and financial livelihood. To be useful, a culture target ideally would include clear land management goals and concrete numbers for sustaining culture and heritage trees for communities throughout southeast Alaska. A culture target would aid in development of tangible actions facilitating current land management goals (such as stated in the Tongass Land Management Plan) while also integrating TEK guidance to safeguard trees long into the future.

Although our "culture target" conceptual model may not fully render data derived from community discussions in a way that is true to Tlingit, Haida, and Tsimshian culture, the model serves to point out implications of sustainable forests on supporting vital cultural lifeways. The Tlingit philosophy of "place and being," emphasizing connections and caretaking promoting "biocultural health," highlights a heightened need to cherish and celebrate threatened resources [73]. Given cultural importance, redcedar has been heralded as a species that should be protected as a keystone species with use focused on maintenance of sustainable cultural ways of life [74]. Interdependence of human communities and the ecosystems sustaining them for thousands of years are central to a new 'ecosystem philosophy' grounded in creative and restorative justice [75].

We acknowledge the limits to our research study including possible incomplete TEK collection and use, non-random participant sampling method, small sample size, and restricted knowledge of forest characteristics accessed to support wood cultural use. Although we strove to record beliefs, values, and practices that include descriptions of

TEK [24], we acknowledge that with our use of western reductionist coding to determine the multiple themes associated with significance and management, we imposed biases on how data was managed and presented. More research and further partnering with southeast Alaska Tribes is needed to better understand the interacting themes associated with cultural use of wood products. While this study, by inclusion of a range of participants that had experience in carving and weaving (>75% of participants), the non-random nature of participant selection did not support population inference from the analysis. Further, better understanding of the forest conditions are needed to sustain culture and heritage activities. Such studies would include assessments of the distribution, density, volume, quality, age, diameter, landscape, microsite, and forest harvest and natural disturbance history of cedar resources, e.g., [14,76–79].

*4.3. Management Suggestions for Sustaining Culture and Heritage*

This study identifies opportunities for land managers to engage and acknowledge ecosystem services associated with cultural wood products. Given the USFS stated mission, "To sustain the health, diversity, and productivity of the Nation's forests and grasslands to meet the needs of present and future generations" [80], one opportunity is providing and improving access to trees for Indigenous cultural purposes. The USFS, for instance, has already granted tribal access to "trees, portions of trees, or forest products for noncommercial traditional and cultural purposes" [81] on a request basis. Yet as one participant noted, given the noncommercial requirement, it is "difficult, or even impossible to make a living wage as an Alaska Native carver in our society given the great expense of purchasing and delivering a large diameter tree." Hence, "culture targets" or synergies with existing timber sales may improve artisan access to raw materials. Another means of access discussed by participants were roads. Most participants in this study believed that although roads increase access to cultural resources for local peoples, the roads are built expressly to extract resources (typically timber). The concern is that if roads are prioritized for widespread timber use rather than local access, roads could effectively decrease community members' access to trees to be used for culture and heritage.

In addition to identification of "culture targets", a systematic analysis of "cultural market values" could provide balance between requirements for sustaining wood products for culture and heritage with timber targets. Such identification, conservation, and restoration of social-ecological systems [82] including use of specific resource may be facilitated by protecting cultural keystone species such as cedar [74]. Further, communication of strategies to preserve and foster wood for culture and heritage are needed.

Our summary of discussions also identified opportunities for collaboration. The southeast Alaska region has developed several recent rich collaborations, including Alaska Tribes and conservation organizations, and the Tongass Advisory Committee. In this study, community members expressed interest in collaborative stewardship with land management to further support culture and heritage activities. Community members frequently described difficulties with current land management policies including insufficient awareness or communication. More collaborative relationships would serve to improve education, communication, and enhance the provision of ecosystem services [83]. Stewardship practice frameworks that have been developed for fish and aquatic resources and ecosystem service frameworks have been heralded by others [84,85]. For example, one study in the state of Oregon found that most survey respondents reported satisfaction with stewardship collaborations [83]. Meaningful community engagement in management decisions can lead to greater participant satisfaction, shared ownership, enhanced trust, strengthened relations among parties, and enhanced community resilience [86].

In this study, several participant-suggested management strategies were articulated to increase supply and access to trees. These included heightened engagement of local artisans and Alaska Tribes in forest planning efforts, roadside delivery of selected trees from timber sales, improvement of the permit process, allocation of certain cedar groves for cultural use only, and allowing bark harvest prior to timber sales. Using

community discussion recommendations (Table 5), we developed an eight-step cycle of collaborative stewardship between native communities and land managers (Figure 5). Development of an adaptive plan of collaborative stewardship would include steps 1 through 5. Implementation of the adaptive plan would include steps 6 to 8. Given shared understanding of wood products for culture and heritage (step 1, along with Figure 4), further discussion of community needs would be described (step 2) and facilitated with representatives from both communities and land managers. The meeting of representatives would enhance understanding of tree resources needed (step 3). Such discussions would also help to compile culture market values leading to a shared culture plan (step 4) and development of culture targets (step 5). A documented culture plan (step 6) and implementation of culture targets would create concrete steps toward sustaining culture and heritage targets (step 7). Update of stewardship targets would occur on a regular basis to adaptively manage community-based stewardship targets (step 8). In alignment with eight-step cycle of collaborative stewardship, our engagement of community members, including Alaska Native youth, helped to foster greater representation and communication of cultural needs (steps 2 and 3).

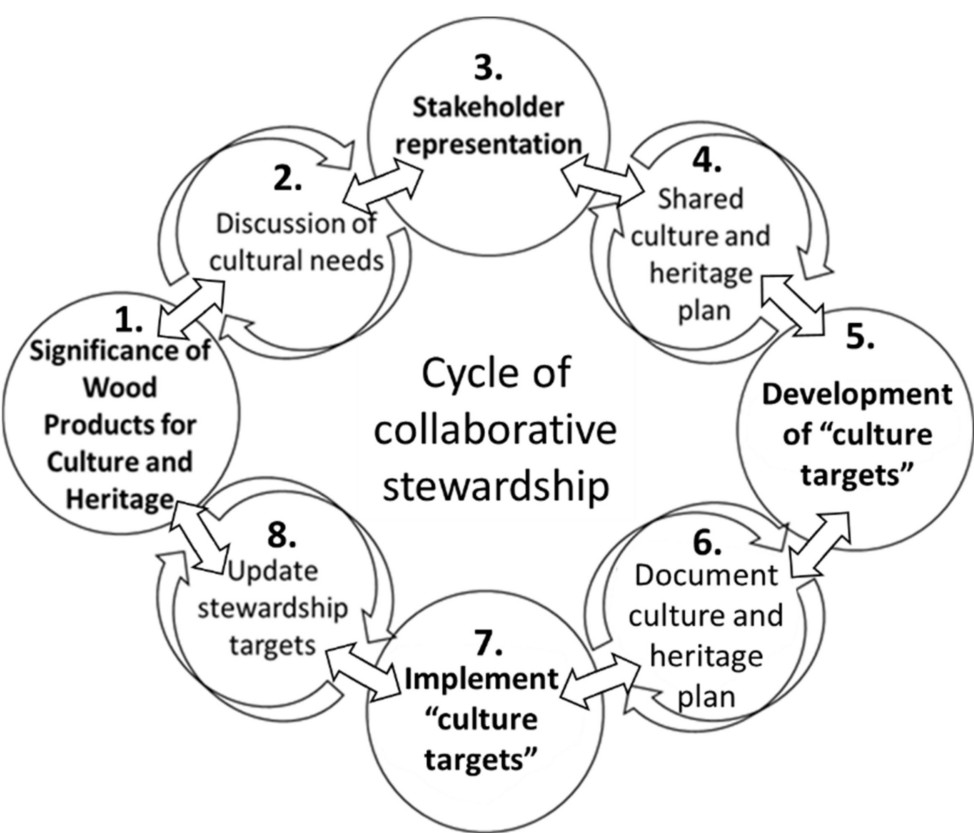

**Figure 5.** Eight-step adaptive cycle of collaborative stewardship between Alaska Native communities and land managers. Steps 1, 3, 5, and 7, in bold type indicate major goals in the stewardship process. Arrows and circular cycles indicate inclusive multi-direction means of communication.

## 5. Conclusions

Participation in culture and heritage activities produces positive social effects, and maintaining those traditions is aided by understanding and incorporating Indigenous perspectives and worldviews. These worldviews can benefit management plans aimed at sustaining culture and heritage practices. This study illuminated common community themes attributed to wood products used for culture and heritage activities such as balance, passing on traditions, cultural identity, positive health outcomes, and community cohesiveness. Cultural revitalization in southeast Alaska relies on the sustained presence

and access to wood resources. Study participants described ways in which participation in cultural activities using wood and bark promoted well-being not only to Alaska Native and non-native community members, but also participants described how well-being was transmitted to travelers from around the world that have engaged with Alaska Native artisans in carving and weaving activities.

Our research indicates Alaska Natives desire greater dialogue with forest managers to secure access to wood and bark products, and to ensure that their descendants, seven generations into the future will be able to harvest these same products for cultural purposes. Alaska Natives have historically managed these resources for sustainability as this land's first stewards, conservationists and land managers. Actively engaging Alaska Natives in contemporary land-use policy development fosters sustainable cultural harvest of wood products in ancient homelands and enhances environmental justice, e.g., [87]. Future research weighing the social benefits of sustaining wood product cultural use in the context of simple market timber harvest values is needed for sustaining culture values in communities. Indigenous scientists will be key partners in advancing these research objectives. In other areas of the world, misalignment of these objectives has resulted in loss of both cultural resources and cultural identity and an increase in stressors, e.g., [88].

In summary, forest management strategies supporting Indigenous lifeways could benefit from a "culture target" as part of the forest planning process and lead to better understanding of "culture market values." Future research could include ways that Alaska Native youth stewardship programs can assist with the preservation of cultural knowledge, traditional skills, and natural resources. Additional work is warranted to address Alaska Native sustainability practices and to identify a means to better incorporate TEK into federal forest management policy.

**Supplementary Materials:** The following are available online at https://www.mdpi.com/1999-4907/12/1/90/s1, Supplementary Material, Part I—ArcGIS Survey 123 survey tool; Supplementary Material, Part II—More information on youth programs; Supplementary Material, Part III—Codes used; Supplementary Material, Part IV—Additional quotes other than the ones used in the text.

**Author Contributions:** Conceptualization, A.J., W.P., G.H., D.-A.H. and J.C.; methodology, A.J., J.C., and A.E.C.; software, A.J. and A.E.C.; validation, L.K.C., A.E.C., A.J., and H.W.; formal analysis, A.J., A.E.C.; investigation, A.J., A.E.C., G.H., D.H., A.L., A.C., and S.R.; data curation, A.E.C.; writing—original draft preparation, A.J., A.E.C., G.H., D.-A.H., L.K.C., H.W., W.P., J.C., A.T., A.L., L.P., A.C., AND S.R.; writing—review and editing, A.J., A.E.C., L.K.C., H.W., W.P., J.C., A.T., G.H., A.J., A.L., L.P., A.C. and S.R.; visualization, A.J. and A.E.C.; supervision, A.J. and L.K.C.; project administration, A.J.; funding acquisition, A.J., J.C., and A.T. All authors have read and agreed to the published version of the manuscript.

**Funding:** This research was funded by a USDA Forest Service, Pacific Northwest Research Station Research for Underserved Communities Fund, 2019 and a USDA Forest Service National Citizen Science grant, 2020.

**Institutional Review Board Statement:** This study did not require ethical approval. Ethical review and approval were waived for this study because "Free and Informed Consent Forms" were utilized and the ArcGIS Survey 123 was developed by the Hoonah Indian Association.

**Informed Consent Statement:** Informed consent was obtained from all subjects involved in the study.

**Data Availability Statement:** Data is not publicly available. Data is located in the individual Alaska Tribal archives as agreed upon by project partners. Data presented in this study, excluding all personally identifiable information, may be available on request from the corresponding author.

**Acknowledgments:** Numerous visionary people including Wayne Price, Lillian Petershoare, Robert Girt, and William Bennett were key in launching this project. Gary Lawton and Pat Tierney assisted in establishing a cedar seedling student experiment and Jennifer Hamblen created the initial education curriculum. We extend our gratitude to the 58 community participants. Specific thanks are extended to the 18 youths and student interns who led conversations including: Andrea Cook, Sienna Reid; Angoon: Brennan Bales, Leonna Demmert, Julian Duncan, Cheyenne Kookesh, Shaylianna

Kookesh, Caleb Jack, Kji Mitchell; Hoonah: Gabrielle Comolli, Theodore Elliot, Dylan Johnson; Kake: Courtney James, Ethan Kadake, Bree Travica, Brandon Ward, Chad Ward. Appreciation is extended to Alexandria Freibott and Ian Johnson who helped create and manage the ArcGIS Survey 123 tool. We thank the reviewers of the discussion questions including Sonia Ibarra, Marina Anderson, Melinda Hernandez-Burke, Katie Riley, and Dennis Nickerson. Both discussions and reviews of the manuscript by Linda Kruger, Elizabeth MacWhinney, Merry Ellefson, and Dede Olson greatly enhanced the quality of this article. Eric Hangartner assisted with graphic design. Finally, thanks are extended to the three anonymous reviewers and *Forests* editors.

**Conflicts of Interest:** The authors declare no conflict of interest. The funders had no role in the design of the study; in the collection, analyses, or interpretation of data; in the writing of the manuscript, or in the decision to publish the results.

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
