# Peer review of "Wood Products for Cultural Uses: Sustaining Native Resilience and Vital Lifeways in Southeast Alaska, USA"

_forests, doi:10.3390/f12010090_

Round 1

Reviewer 1 Report

This is a wonderful paper, not only for critical topic and highly relevant and informative research results, but also for the rich inclusive and participatory methods used in the project. Bravissimo. I look forward to using (and citing!) some of the ideas and approach used here in our own work with Tribal populations in the Himalaya. And I very much look forward to seeing the ArcGIS StoryMap. The authors take great care in identifying clear ways to improve the Indigenous people's access to these critical resources (trees, bark).

The content is quite good, but the paper needs a careful edit for punctuation, grammar, and visual design. There are several conventions that are not followed and many small errors. I have listed some here.

  • The period goes inside the quotation marks
  • No quotation marks are necessary on widely recognized terms, such as keystone species (line 761) or socio-ecological systems (line 801)
  • There are many stray double spaces throughout the text
  • The use of bullet points for the quotes in the text body is also distracting. The convention is simply to indent the quote.
  • Improper and distracting use of ”n=” throughout the paper. The symbol n= refers to a sample size, but the authors use it to refer to the number of respondents with a certain answer or characteristic.

Small problems, by line

Punctuation problems (such as missing comma, unnecessary comma, apostrophe, etc.)  on the following lines: 104, 106, 116, 126, 137, 138, 255, 341, 378, 423, 571, 759, 826

Line 262 – unclear the use of the word ”amenity”

Lines 376, 389 – stray text?

Line 394 – a word missing?

Line 422 – use of ”style” twice is confusing

After line 467 a close parenthasis is missing

Line 470-472 – confusing. Who are the ”few”? How many is a ”few”?

Line 535 – stray text?

Line 632 – stray word ”as”?

Line 740 – delete ” the and”

Lines 741-743 – awkward. Perhaps delete the ”Whereas … project” and say, ”Five quantifiable components of cultural targets  suggested … ”

Lines 765-767 – unclear meaning of this sentence.

Lines 809, 810 – double use of ”is”

Line 853 – use of the semicolon is incorrect and leaves the second half as a fragment only

Lines 856-858 – check the grammar

Other problems

The term ”biologic” is odd; I have not encountered it before -- more appropriate would be ”biological”

The specific objectives stated in lines 174-178 do not correspond to the objectives stated in lines 283-291. And more ”project goals” are stated in lines 208-211 – should these be stated above in the Introduction?

Tables

First principle: minimize the use of repeated text or characters (eg, in Table 1,, the many n=). Second principle: left justify table entries. Third principle: no bullets.

Table 1 is a visual nightmare (sorry, but it is). Take care of the repeated information:  the bazillion ”n=”, the Y, the N, the TOACM, the age groups). Perhaps consider a graph instead. Perhaps for Type, use five columns in the heading, one for teach type (TMACO), then put the number of respondents in the table. Same with the Y/N.

Table 2 is difficult to read. I suggest the following:

  • correct the use of ”n=” – change to simply putting the number of respondents in each category in parentheses, e.g. (26), and in the caption say ”the number of respondents is in parentheses” as you do in Table 4 (but ditch the ”n=”)
  • left justify the significance (not center), and line up the Type and Significance to the top of each line.

Table 3. Same comments, and do not use bullets (they are highly displeasing to the eye) – just line stuff up on the left.

Caption to Figure 5 does not need to repeat the text in the figure since the steps are explained in the text.

Author Response

Hello Reviewer One,

We have addressed all your comments, and consequently, the manuscript has been improved substantially. All of your comments and suggestions are addressed as follows in italics:

This is a wonderful paper, not only for critical topic and highly relevant and informative research results, but also for the rich inclusive and participatory methods used in the project. Bravissimo. I look forward to using (and citing!) some of the ideas and approach used here in our own work with Tribal populations in the Himalaya. And I very much look forward to seeing the ArcGIS StoryMap. The authors take great care in identifying clear ways to improve the Indigenous people's access to these critical resources (trees, bark). 

We are pleased that you appreciate our approach and are very glad that it will have application to your own work!  We now have a link to the ArcGIS StoryMap in the manuscript that will become “live” once it is formally approved.

The content is quite good, but the paper needs a careful edit for punctuation, grammar, and visual design. There are several conventions that are not followed and many small errors. I have listed some here.

We have addressed all of your comments.  The manuscript has been improved significantly due to your careful review.  Thank you very much for your diligent attention.

  • The period goes inside the quotation marks. Thank you. We worked to address this issue.
  • No quotation marks are necessary on widely recognized terms, such as keystone species (line 761) or socio-ecological systems (line 801). Quotation marks were removed.
  • There are many stray double spaces throughout the text. Thank you for pointing this out. We strove to remove stray double spaces.
  • The use of bullet points for the quotes in the text body is also distracting. The convention is simply to indent the quote. All bullet points were removed.
  • Improper and distracting use of ”n=” throughout the paper. The symbol n= refers to a sample size, but the authors use it to refer to the number of respondents with a certain answer or characteristic. Thank you for making this suggestion.  We removed the n= throughout which improves the document.

Small problems, by line

Punctuation problems (such as missing comma, unnecessary comma, apostrophe, etc.)  on the following lines: 104, 106, 116, 126, 137, 138, 255, 341, 378, 423, 571, 759, 826.  Aimed to make all changes. When unsure, sentences were simplified.

Line 262 – unclear the use of the word ”amenity”. It was changed to read to “maintain participant confidentiality.”

Lines 376, 389 – stray text?  Removed the word “response”, 389 not sure here, but footnotes were organized under table.

Line 394 – a word missing?  Clarified the sentence by adding “historical trauma” after “recovery from”.

Line 422 – use of ”style” twice is confusing. Simplified sentences and removed one “style”.

After line 467 a close parenthesis is missing. Added.

Line 470-472 – confusing. Who are the ”few”? How many is a ”few”?  It is now clarified as being five with three associated with forest harvest and mills

Line 535 – stray text? Added “Other” to next line.

Line 632 – stray word ”as”?  Removed.

Line 740 – delete ” the and”  Removed.

Lines 741-743 – awkward. Perhaps delete the ”Whereas … project” and say, ”Five quantifiable components of cultural targets  suggested … ”  Change made.

Lines 765-767 – unclear meaning of this sentence. Rewrote sentence to clarify.

Lines 809, 810 – double use of ”is”. Removed second “is”

Line 853 – use of the semicolon is incorrect and leaves the second half as a fragment only. Rewrote sentence.

Lines 856-858 – check the grammar. Added “described how” after “but also”.

Other problems

The term ”biologic” is odd; I have not encountered it before -- more appropriate would be ”biological” Changed.to biological.  

The specific objectives stated in lines 174-178 do not correspond to the objectives stated in lines 283-291. And more ”project goals” are stated in lines 208-211 – should these be stated above in the Introduction?

Spiritual, health, tourism, and income are now included as the array of cultural values measured as a project objective. Although the seedling experiment was discussed and initiated following community meetings, as described in the community discussion section (2.1), the seedling experiment was not part of this focused research study, so the seedling experiment is not described in the introduction.

Tables

First principle: minimize the use of repeated text or characters (eg, in Table 1,, the many n=). Second principle: left justify table entries. Third principle: no bullets.  Thank you.  All changes made

Table 1 is a visual nightmare (sorry, but it is). Take care of the repeated information:  the bazillion ”n=”, the Y, the N, the TOACM, the age groups). Perhaps consider a graph instead. Perhaps for Type, use five columns in the heading, one for teach type (TMACO), then put the number of respondents in the table. Same with the Y/N. 

The table has been cleaned up and simplified per your suggestions.  We tried to make a bar plot, but simplifying the Table was clearer. I hope you like it much better!

Table 2 is difficult to read. I suggest the following:

  • correct the use of ”n=” – change to simply putting the number of respondents in each category in parentheses, e.g. (26), and in the caption say ”the number of respondents is in parentheses” as you do in Table 4 (but ditch the ”n=”)
  • left justify the significance (not center), and line up the Type and Significance to the top of each line. Done. Thank you for these suggestions.

Table 3. Same comments, and do not use bullets (they are highly displeasing to the eye) – just line stuff up on the left.  Bullets removed.

Caption to Figure 5 does not need to repeat the text in the figure since the steps are explained in the text. Details of the caption were removed.

Thank you again for your thorough review. 

Reviewer 2 Report

Review of “Wood Products for Cultural Uses: Sustaining Native Resilience and Vital Lifeways in Southeast Alaska, USA”

The authors describe in their article significance of wood products to rural communities in Southeast Alaska. Based on results from 58 discussions across Alaska communities, authors highlighted the cultural importance of forest products. Also obtained results are very interesting and in my opinion the article will be interesting for readers.

The statement on page 14 (line 506-507) „Bureaucratic processes: Participants shared frustrations with the current administrative permit process (to secure trees for culture use on USFS lands) policies“ is true and common for several countries mainly in Europe.

Discussion of the results is very good and also possible management strategies are mentioned

In general, the paper is very well balanced and I have no further comments.

Author Response

Hello Reviewer Two,

Your comments are addressed as follows in italics:

The authors describe in their article significance of wood products to rural communities in Southeast Alaska. Based on results from 58 discussions across Alaska communities, authors highlighted the cultural importance of forest products. Also obtained results are very interesting and in my opinion the article will be interesting for readers.

Thank you very much for your feedback.

The statement on page 14 (line 506-507) „Bureaucratic processes: Participants shared frustrations with the current administrative permit process (to secure trees for culture use on USFS lands) policies“ is true and common for several countries mainly in Europe.

We aim to help with these frustrations, if possible.  

Discussion of the results is very good and also possible management strategies are mentioned

This is wonderful to hear.  We believe the revised manuscript is improved.

In general, the paper is very well balanced and I have no further comments.

Thank you again for your review.

Reviewer 3 Report

Dear Authors,

I have only minor editorial comments- please fill the official statements at the end of the paper;

otherwise, congratulations on interesting work,

Reviewer 

Author Response

Dear Reviewer Three,

We have addressed all the reviewer comments, and consequently, the manuscript has been improved substantially. 

All comments are addressed as follows in italics:

I have only minor editorial comments- please fill the official statements at the end of the paper; otherwise, congratulations on interesting work,

I have corresponded with Ms. Whitney Wu, Associate Editor to get a better understanding of any specific recommendations from reviewer three that would enhance the introduction and discussion.  We have worked to reconcile all suggestions from reviewer one and believe that these additions have helped with the introduction, discussion, and overall quality of our paper.  We have filled out the official statements at the end of the paper. 

Thank you for your review.